# Development of a Macrophage-Related Risk Model for Metastatic Melanoma

**DOI:** 10.3390/ijms241813752

**Published:** 2023-09-06

**Authors:** Zhaoxiang Li, Xinyuan Zhang, Quanxin Jin, Qi Zhang, Qi Yue, Manabu Fujimoto, Guihua Jin

**Affiliations:** 1Department of Immunology and Pathogenic Biology, Yanbian University Medical College, Yanji 133002, China; 2195113071@ybu.edu.cn (Z.L.); 0000008740@ybu.edu.cn (X.Z.); jinqx@ybu.edu.cn (Q.J.); 2022001076@ybu.edu.cn (Q.Z.); 2020010505@ybu.edu.cn (Q.Y.); 2Laboratory of Cutaneous Immunology, Osaka University Immunology Frontier Research Center, Department of Dermatology, Graduate School of Medicine, Osaka University, Osaka 565-0871, Japan; mfujimoto@derma.med.osaka-u.ac.jp

**Keywords:** melanoma, tumor metastasis, biological makers, tumor immune microenvironment, M1/M2 macrophage

## Abstract

As a metastasis-prone malignancy, the metastatic form and location of melanoma seriously affect its prognosis. Although effective surgical methods and targeted drugs are available to enable the treatment of carcinoma in situ, for metastatic tumors, the diagnosis, prognosis assessment and development of immunotherapy are still pending. This study aims to integrate multiple bioinformatics approaches to identify immune-related molecular targets viable for the treatment and prognostic assessment of metastatic melanoma, thus providing new strategies for its use as an immunotherapy. Immunoinfiltration analysis revealed that M1-type macrophages have significant infiltration differences in melanoma development and metastasis. In total, 349 genes differentially expressed in M1-type macrophages and M2-type macrophages were extracted from the MSigDB database. Then we derived an intersection of these genes and 1111 melanoma metastasis-related genes from the GEO database, and 31 intersected genes identified as melanoma macrophage immunomarkers (MMIMs) were obtained. Based on MMIMs, a risk model was constructed using the Lasso algorithm and regression analysis, which contained 10 genes (NMI, SNTB2, SLC1A4, PDE4B, CLEC2B, IFI27, COL1A2, MAF, LAMP3 and CCDC69). Patients with high+ risk scores calculated via the model have low levels of infiltration by CD8^+^ T cells and macrophages, which implies a poor prognosis for patients with metastatic cancer. DCA decision and nomogram curves verify the high sensitivity and specificity of this model for metastatic cancer patients. In addition, 28 miRNAs, 90 transcription factors and 29 potential drugs were predicted by targeting the 10 MMIMs derived from this model. Overall, we developed and validated immune-related prognostic models, which accurately reflected the prognostic and immune infiltration characteristics of patients with melanoma metastasis. The 10 MMIMs may also be prospective targets for immunotherapy.

## 1. Introduction

Cutaneous melanoma, which is a highly aggressive form of skin cancer arising from pigment-producing cells located in the epidermis, is traditionally viewed as one of the most metastatic malignancies. As melanoma metastasis spreads to vital organs, such as brain, lung and liver, it often causes the eventual death of patients. Sophisticated surgery, radiotherapy, newer immunotherapeutic and molecularly targeted agents have led to a substantial increase in metastatic cancer patient survival. Given the inevitable development of resistance and tumor recurrence, a large subset of patients fail to benefit from current anti-PD-1 or anti-CTLA-4 therapies.

With an incredibly high genomic mutational load, malignant melanoma shows immunogenic plasticity, which leads to its immune escape. This outcome occurs due to the lower immunogenic phenotype acquired in melanoma cells and this cancer’s ability to inhibit anti-tumor immune cells in the tumor microenvironment (TME). Many factors contributing to melanoma immune escape have been elucidated, including a low MITF/high c-Jun cell state that promotes the establishment of a pro-inflammatory and myeloid-rich tumor microenvironment, as well as interconnected neutrophil inflammation and melanoma cell plasticity, leading to the regional and distant vascular metastasis of melanoma [1,2]. However, the view that the immune microenvironment alters the melanoma plasticity driving its metastasis remains unconfirmed, and this issue has been a major obstacle to the development of effective treatment and diagnosis.

As tumor cells acquire different immunogenicity characteristics, they secrete immunomodulatory molecules to alter TME, which influences the activation of immune cells or the composition of intratumor immune invasive cells. Independent studies of melanoma and other solid malignancies have identified many key congenital and adaptive subpopulations within antitumor activity [3,4]. These tumor-associated antigens are divided into the following groups: lineage-specific markers that are overexpressed in melanoma cells, such as melanoma antigen recognized by T cells; tyrosinase and glycoprotein 100; cancer-testis antigens; neoantigens; and antigens originating from somatic mutations [2]. However, due to the high plasticity of the cellular state during melanoma metastasis, integrating these immune targets becomes more challenging.

The most optimal therapeutic strategy is considered to involve integrating most of the immune subpopulations without causing significant toxicity. In our previous study, we found that CD8^+^ CTL cells were effective at inducing the apoptosis of melanoma cells in mice with metastatic cancer, as inhibition of the immunosuppressive effect of the Treg cell can impress melanoma in situ [5,6]. It has been demonstrated that M1 macrophages expressed high levels of CD86, MHC II and B7 and mediated anti-tumor effects by secreting high-level cytokines (such as IL-12, IL-23, IL-1 and IL-6) and chemokines (MCP-1, CCL3, CXCL9, CXCL10, etc.) [7]. Susek et al. indicated that M1 macrophages recruited CD8^+^ T cells and NK cells by secreting chemokines, such as CXCL9 and CXCL10, to exert anti-tumor effects in the early stage of non-small cell lung cancer and ovarian cancer [8]. In contrast, M2 macrophages have different phenotypes and functions. Representative M2 markers include ARG1, CD163 and CD206, and the infiltration of M1 within the tumor is closely related to the poor prognosis of the tumor [9].

However, it is still unclear which immune cell population is more dominant during melanoma metastasis and how researchers can develop diagnostic and therapeutic approaches to treat metastatic melanoma based on its immune-related targets. In this study, we used bioinformatic approaches to analyze the characteristics of immune cell infiltration in metastatic melanoma, and we determined that M1-type macrophages were highly associated with melanoma metastasis. Then, we screened for differentially expressed genes (DEGs) that were strongly associated with melanoma metastasis via bioinformatic analysis of a large-scale sequencing database. Next, the DEGs with M1-type macrophages were further examined, and 31 macrophage immune marker molecules for metastatic melanoma (MMIMs) were obtained by taking intersections of melanoma metastasis-related genes. Notably, we constructed an immune-related prognostic model to predict metastatic melanoma by integrating the MMIMs of melanoma. In addition, we predicted the potential regulatory mechanisms and targeting agents of these molecules. The aim of this study is to construct a new multiple dimensional model that can effectively predict the prognosis of metastatic patients, as well as to provide the potential targets of clinical treatment of metastatic melanoma.

## 2. Results

### 2.1. Identification of Metastasis-Associated Melanoma DEGs

The specific DEGs screening process is shown in Figure 1. Three gene expression dataset profiles (GSE7929, GSE7956 and GSE8401) were retrieved from the GEO database. The detailed information of the data is shown in Table 1. According to the criteria of *p* < 0.05 and |logFC| ≥ 1.0, 511 differently expressed genes (DEGs) were obtained from GSE7929, which contained 217 upregulated genes and 294 downregulated genes. The 56 DEGs were obtained when the gene chip GSE7956 was filtered, containing 21 upregulated genes and 35 downregulated genes. In the GSE26440 dataset, 674 DEGs were obtained, 477 of which were upregulated and 197 of which were downregulated. All DEGs were derived from the comparisons between transferred and untransferred samples. The heat and volcano maps (Figure 2A–C) all indicated large differences in DEGs between the three groups of metastatic and non-metastatic melanomas. PCA plots of the data derived from the three sample groups (Figure 2D) showed differences in the expression patterns between the metastatic and non-metastatic melanoma groups. A total of 1111 DEGs in these three groups were identified as melanoma metastasis-associated genes.

### 2.2. The Roles Played by M1 Macrophages in Melanoma Development and Metastasis

Elucidating the interplay between cancer and the host immune system is critical to identifying key pathogenic molecules, improving drug sensitivity and developing new therapeutic strategies. To investigate the translation of the metastasis tumor immune microenvironment to melanoma, we estimated the abundance of immune cells in primary melanoma and melanoma metastasis based on RNA-seq data using TIMER 2.0 software. We extracted transcriptomic data in situ from normal skin compared to melanoma in situ in carcinoma in GSE114445 to analyze the differences in immune infiltration between the two groups. The immune infiltration results showed that resting NK cells, M1 macrophages, activated mast cells and neutrophils have significant differences between two groups (Figure 3A). This finding suggested that these cells may be involved in the initiation of melanoma.

We extracted the transcriptional information of metastatic cancers and cancers in situ in GSE8401 and analyzed their immune cell infiltration. We found that B cell memory, CD4^+^ T cell (resting and activated), M1 Macrophage and Myeloid dendritic cell showed different abundances of infiltration through the process taking place in highly metastatic melanoma (Figure 3B). Interestingly, M1-type macrophages and mast cells consistently infiltrated in different abundances in normal skin compared to primary melanoma, and similar results were found in primary melanoma compared to metastatic carcinoma, suggesting that these two types of immune cells may offer meaningful clues regarding the progression process of melanoma. Given the significance of macrophages in the development of tumors, we determined that M1-type macrophages were worth studying regarding the metastasis and occurrence of melanoma.

### 2.3. Functional Analysis of Key Genes and Identification of MMIMs

We identified three groups of DEGs involved in melanoma development and metastasis. To understand the biological functions and signaling pathways behind these DEGs, we took advantage of corresponding KEGG and GO enrichment analysis, finding that the differential genes extracted from GSE7929, GSE7956 and GSE8401 were mainly enriched in extracellular matrix tissue, in extracellular structural tissue, in collagen-containing extracellular matrix, via the PI3K-AKT signaling pathway, through ECM–receptor interactions and via other pathways (Figure 4A–C).

Meanwhile, immune infiltration studies have shown that M1-type macrophages play an important role in the development of metastatic melanoma. In previous studies, M1-type macrophages were usually considered to be tumor-killing macrophages due to their anti-tumor properties. In contrast, M2-type macrophages promote tumor growth and metastasis and are associated with the poor prognosis of a tumor [9,10]. DEGs in M1- and M2-type macrophage polarization may be the most important cause of melanoma metastasis. We extracted DEGs of M1- and M2-type macrophages from a database, intersecting them with 1111 melanoma metastasis-related genes to obtain 31 macrophage immunomarker MMIMs (Figure 4D). We also found that these MMIMs were mainly associated with specific pathways, such as cell–cell adhesion, positive regulation of Th1 and Th2 cell differentiation, positive regulation of CD4 -positivity and the activation of α-β T cells (Figure 4E). These pathways are related to the metastasis and immune regulation of melanoma.

### 2.4. Construction of a Risk Model for Macrophage Immunomarker Molecules of MMIMs

Based on 31 obtained MMIMs, we extracted the same 31 MMIMs from 167 melanoma metastases patients to apply final LASSO regression modeling in order to validate its role in clinical settings (Figure 5A). Model 1 was optimized via tenfold cross-validation when λ = 0.055, containing 10 key genes as variables (Figure 5B). Based on the expression of the 10 model genes, a risk score was obtained: the patient score for macrophage immune-related molecular risk = (−0.052 × NMI) + (−0.0659 × SNBT2) + (0.0394 × SLC1A4) + (−0.0535 × PDE4B) + (−0.1921 × CLEC2B) + (−0.0823 × IFI27) + (0.1018 × COL1A2) + (0.1031 × MAF) + (−0.0338 × LAMP3) + (−0.0512 × CCDC69). Based on this assessment model, we divided the patients into high- and low-risk groups, finding that risk scores, survival times and survival status profiles in the selected dataset were significantly correlated (Figure 5C). The associated prognostic analysis curves and diagnostic curves suggested that this model had promising diagnostic significance for the TCGA database of 167 patients with metastatic melanoma (Figure 4E, AUC = 0.802), while the prognoses of low-risk patients distinguished via this model were significantly better than those of high-risk patients (Figure 5D).

### 2.5. Prognostic Models Constructed Based on MMIMs Molecules Have Better Prognostic Effect and Specificity in Melanoma Metastatic Cancer

To further compare the diagnostic and prognostic values of the 31 MMIMs, we extracted 471 melanoma patients from the same TCGA database (without differentiating between metastases and non-metastases) and constructed a risk model using the Lasso method. Model 2 was optimized via tenfold cross-validation when λ = 0.0422, containing 10 key genes variables. A risk score assessment model (model 2) was obtained based on the expression of 11 model genes (Figure 6A). The risk score of the patient = (−0.1179 × NMI) + (−0.0025 × S100B) + (−0.0549 × PDE4B) + (−0.1254 × CLEC2B) + (−0.0333 × IFI27) + (0.0472 × COL1A2) + (0.0089 × SMPD3) + (0.0466 × ANK3) + (0.021 × MAF) + (−0.0926 × LAMP3) + (−0.0015 × CCDC69).The prognosis of the low-risk population obtained using the Model 2 grouping was also better than that of the high-risk population (Figure 5B), though its specificity was unsatisfactory, having an AUC < 0.75 (Figure 6C). However, we found that the diagnostic efficacy increased with increasing prediction years, which may be associated with distant cancer progression leading to metastasis.

To compare the application value of these two models for patients with metastasis, we constructed DCA decision curves for Model 1 and Model 2. Model 1 outperformed Model 2 in the decision curves for overall survival after 1, 2 and 3 years (Figure 6D–F).

### 2.6. Clinical Evaluation of the Model

To further evaluate the value of the model in clinical application, we performed univariate regression analysis and multifactor regression analysis of the 10 genes in Model 1 (Figure 7A). CLEC2B and IFI27PDE4B were used as independent risk factors of melanoma metastasis. Using the risk score and other clinicopathological factors, including clinical M-stage and TNM grading, a nomogram was established to accurately estimate the probability of survival after 1, 2 and 3 years in patients with metastatic melanoma cancer (Figure 7B). Its calibration curve analysis also showed that the predicted 1-, 2- and 3-year survival times were consistent with the actual survival times (Figure 7C). The expression of these last 10 corresponding MMIMs in human macrophages is shown in Appendix A.

### 2.7. Analysis of Immune Correlation in Targeted Model

Correlation analysis based on a model targeting six immune cells in patients with metastatic melanoma in TCGA (Model 1) showed that the risk score of the model correlated with the expression of macrophages (Spearman’s coefficient = −0.43, *p* < 0.01) and CD8^+^ T cells (Spearman’s coefficient = −0.46, *p* < 0.01) (Figure 8A). This result suggests that model scores were negatively correlated with macrophage and CD8^+^ T cell infiltration, and the infiltration of these two types of cells was associated with the metastasis of melanoma.

### 2.8. The TF-Gene-miRNA Co-Regulatory Network

In previous studies, the regulation mechanism of microRNAs (miRNAs) and related transcription factors (TFs) of relevant diseases were often regarded as important links in terms of mediating the occurrence and development of these diseases. We analyzed the miRNA-gene-TF interactions and detected 90 central regulatory TFs and 228 miRNAs via the network based on the topological parameters, suggesting that these TFs and miRNAs may be involved in the regulation of these genes, which may affect melanoma metastasis (Figure 8B). The top 10 miRNAs and TFs results for TF-gene-miRNA were provided in Table 2, Table 3 and Table 4.

### 2.9. Protein-Drug Interactions Predict Potential Drugs

To identify potential drugs that may influence the proteins encoded by the core genes, we performed a protein-drug interaction analysis. In total, 10 genes of Model 1 were associated with these drugs. A total of 29 potential drugs was obtained (shown in Table 5). These drugs were categorized into different therapeutic classes, including antineoplastic and immunomodulating agents, immunosuppressive agents, antiplatelet agents, and anti-infective agents, and further classified based on their progress in the therapeutic pipeline. Moreover, etanercept and adalimumab have been reported to treat melanoma, while the rest of these drugs have not been directly used to treat melanoma, as determined by searching the ClinicalTrials.gov registry.

## 3. Discussion

As a highly malignant and metastatic tumor, melanoma has an extremely poor prognosis once metastasized. Globally, the accurate prediction of overall survival in metastatic melanoma is important for treatment selection and prognosis improvement. To date, there are no reliable and valid biomarkers that precisely predict the survival of patients with metastatic melanoma [10]. Therefore, there is an urgent need to identify robust biomarkers and predictive models to predict the outcomes of melanoma metastasis.

In the current study, 1111 differentially expressed metastasis-related genes were screened from 102 metastatic melanoma samples (patients and cells) and 52 non-metastatic melanoma samples (patients and cells) based on the analysis of the GEO dataset. Based on the results of GO and KEGG enrichment analyses, the above genes were mainly associated with metastasis (e.g., extracellular matrix organization, extracellular structural organization and collagen-containing extracellular matrix signaling pathways). Interactions between melanoma and immune cells in the tumor microenvironment contribute to melanoma metastasis [11]. Immune cell infiltration analysis in primary melanoma and melanoma metastasis revealed the involvement of M1-type macrophages in melanoma development and metastasis. Based on the opposite roles played by M1 and M2 in melanoma [12,13], 349 differentially expressed genes were extracted from M1- and M2-type macrophages, 31 of which were melanoma metastasis-related genes. To explore the prognostic roles played by these 31 MMIMs in patients with metastatic melanoma cancer, a LASSO regression model consisting of 10 MMIMs was constructed in 167 patients with metastatic melanoma cancer in the TCGA database, which had a high prognostic value for the prognosis of metastatic melanoma cancer.

The 10 MMIMs were NMI, SNTB2, SLC1A4, PDE4B, CLEC2B, IFI27, COL1A2, MAF, LAMP3 and CCDC69. N-myc and STAT interactor (NMI) are N-Myc interacting factors that enhance the IFN-c-induced transcriptional activity of the transcriptional cofactor [14]. The overexpression of NMI inhibits Wnt/β-catenin signaling, reducing melanoma invasion and metastasis [15]. In other tumors, NMI inhibits COX-2 expression by suppressing p300-mediated p50/p65 NF-κB acetylation to suppress tumor growth [16]. Phosphodiesterase 4B (PDE4B) can directly regulate PDE secretion. PDE is capable of regulating cyclic AMP (cAMP)/PKA activity and regulates the growth of human malignant melanoma cells [17,18]. High expression of PDE in melanoma cells inhibits cAMP pathway transduction, and the inhibition of PDE expression activates the cAMP pathway to inhibit proliferation, induce apoptosis and inhibit the migration ability of Ras mutant melanoma cells [17,19]. CLECB is called C-type lectin domain family 2 member B. Its expression product, which is known as C-type lectin-like receptor 2 (CLEC-2), is a sugar-binding protein. CLEC-2 and its endogenous ligand podoplanin (PDPN) promote hematogenous cancer metastasis and cancer-associated thrombosis [20]. In vivo, the inhibition of CLEC-2 expression was effective at suppressing the microvessel density of melanoma and reducing the interaction with PDPN to block malignant melanoma growth and lung metastasis [21,22]. Interferon alpha inducible protein 27 (IFI27) mediated the activation of the type I interferon-related pathway [23]. Using cetirizine in advanced immunotherapy for melanoma patients induces M1 macrophage polarization through the IFI27-related INF-γ pathway [24]. As subset of the stromal collagen gene, COL1A2 is considered to be a potential molecule marker of melanoma, as it is frequently affected by melanoma methylation [25]. COL1A2 is also a fibroblast-specific promoter that affects melanoma angiogenesis and metastasis via the influention of the connective tissue growth factor (CCN2)/CAF pathway [26]. The MAF called MAF bZIP transcription factor can negatively regulate transcription by RNA polymerase II. The expression of MAF in antitumor CD8^+^ T cells contributes to their polarization toward a depleted phenotype, and TGF-β and IL-6 are able to induce MAF expression in CD8^+^ T cells in vitro and in vivo, while tumor-specific CD8^+^ T cells lacking MAF tend to eliminate melanoma cells [27,28]. Syntrophin beta 2 (SNTB2) is expressed both inside of the cell and at the plasma membrane, as it is involved in the constitution of focal adhesion. SLC1A4 is called solute carrier family 1 member 4, and it is capable of a wide range of transporter activities, such as amino acid, L-alanine and L-aspartate. Lysosome-associated membrane protein 3 (LAMP3) is involved in the regulation of interferon-alpha and adaptive immune responses. CCDC69 is called coiled-coil domain containing 69, and it participates in microtubule binding. As for the immunomodulatory mechanism of SNTB2, SLC1A4, LAMP3 and CCDC69 in metastatic melanoma or macrophages, it was not specifically elucidated, though it is still a promising research direction.

Recently, gene models constructed using aberrant mRNAs revealed their great potential to perform prognostic prediction of melanoma, thus attracting widespread attention. For example, Chen et al. constructed a nine-iron-death-associated gene prognostic model, which showed great performance in terms of predicting melanoma prognosis [29]. Another study constructed an immunogenomic signature prognostic model based on gene expression data derived from TCGA to predict overall survival in melanoma [30]. A recent study used a six-gene model-based prognostic approach to treat melanoma patients [31]. However, due to the limitation number of samples, as well as lack of comprehensively exploration of the relationship between specific immune cell-related genes and the prognosis of metastatic melanoma, these models need to be further completed. Compared to the previous studies, our results have several advantages: (1) We identified immune cell populations that significantly differ between primary melanoma and melanoma metastasis, which allowed us to target and analyze immune-related genes. To the best of our knowledge, this study was the first study to explore the association between a large number of differentially expressed genes in M1- and M2-type macrophages and metastasis and investigate the relationship between related genes and prognosis. (2) We developed a new prognostic model using immunity-related genes to detect metastatic melanoma, which was associated with differentially expressed genes in M1- and M2-type macrophages. This prognostic model showed outstanding survival prediction performance in the TCGA database. (3) To perform an in-depth analysis, genes associated with the model, model scores and clinical indicators of melanoma were incorporated to construct Norman diagrams to carry out prognostic assessment of patients with metastatic melanoma. (4) A TF-gene-miRNA regulatory network was constructed based on the 10 genes used in this model, and we predicted potential target drugs to guide subsequent mechanistic exploration and treatment. All of the above results revealed that the prognostic model of DEGs in M1- and M2-type macrophages can be used as a valid marker of the prognostic prediction of metastatic melanoma to guide clinical treatment.

To verify the correlation between the model and immune cells, we analyzed the correlation between immune cells and the model in 174 patients with metastatic melanoma. Low levels of macrophage and CD8^+^ T cells infiltration were shown in high-risk patients. Macrophages and CD8^+^ T cells were negatively correlated with the MMIM-related prognostic model, suggesting that this model may serve as a predictor of increased immune cell infiltration of macrophages and CD8^+^ T cells, which was consistent with our analysis of immune infiltration performed within the sample. Tumor-associated macrophages are mostly recruited by CCL2 released from tumors and their stroma [32]. Existing views suggest that macrophages are broadly divided into two types involved in tumor progression-mediated invasion: anti-tumor M1- and M2-type macrophages. M1-type macrophages are involved in the tumor immune process of melanoma by secreting cytokines, such as IL-1β, TNF-α, IL-12 and IL-18 [13,33,34]. In contrast, M2-type macrophages are activated by Th2 cells and anti-inflammatory stimuli, such as IL-4, -10, -13 and TGF-β, or the monocyte colony-stimulating factor [35]. Consistent with our immune infiltration analysis, M1-type macrophages, which are predominant in carcinoma in situ, were replaced by M2-type macrophages in metastatic carcinoma. It is unclear why the balance between M1- and M2-type macrophages shifts in favor of the M2 type in tumors, but available studies seem to suggest that inducible nitric oxide synthase, IL-4 and IL-10 can induce a shift from M1- to M2-type macrophages [11,36]. The main cause of tumor cell metastasis is the degradation and damage of the basement membrane of endothelial cells in tumor tissues [37]. M2-type macrophages promote the metastasis of tumor cells by secreting matrix metalloproteinases (MMPs), serine proteases and histone proteases to destroy the stromal membrane of endothelial cells and break down the extracellular matrix [38,39]. In contrast, M1-type macrophages kill tumor cells through slow cytotoxic effects mediated by tumor-killing molecules, such as ROS, NO and tumor-associated antibody-dependent cytotoxicity (ADCC) [37,40]. Considering the high plasticity of malignant melanoma and the roles played by M1- and M2-type macrophages, integrating these tumor-associated antigens and the DEGs of M1- and M2-type macrophages are long-term tasks for immunotherapy targeting metastatic melanoma. And the identified MMIMs will be important molecules in terms of probing the phenotypic transformation of macrophages in melanoma metastatic cancer. This study highlights the clinical significance of MMIMs in patients with metastatic melanoma cancer while revealing the correlation between melanoma metastatic cancer and macrophages. However, several limitations of this study remain to be solved, such as whether the validity of the prognostic model of MMIMs should be experimentally determined in a large number of metastatic melanoma samples, and the biological functions of the 10 MMIMs in metastatic melanoma and macrophages need to be further investigated via a series of experiments. Our next research direction will be to elucidate the more definite molecular mechanisms of these MMIMs in melanoma and develop targeted immunotherapies.

## 4. Materials and Methods

### 4.1. Data Source

Three microarray datasets (GSE7929, GSE7956 and GSE8401) of metastatic and non-metastatic melanoma were collected from the GEO database (https://www.ncbi.nlm.nih.gov/geo/ (accessed on 26 April 2023)). The transcriptome expression profiles of normal skin and melanoma in situ were also extracted from the GEO database (GSE114445) and used to detect changes in the immune microenvironment during melanoma development. Differentially expressed genes of M1- and M2-type macrophages were extracted from the MSigDB database (http://www.gsea-msigdb.org/gsea/msigdb/human/search.jsp, accessed on 28 April 2023). Transcriptomic and clinical information related to melanoma patients was collected from the TCGA database (https://tcgagolf.com/ (accessed on 28 April 2023)) to perform prognostic model construction and validation. All files were freely downloaded via the Internet. The details of the gene expression profiles used in this analysis are presented in Table 1.

### 4.2. Sample Immunoinfiltration Analysis

To find the samples’ immunoinfiltration differentiation, we investigated the condition of immunoinfiltration of primary cancer and metastasis in GSE8401 via TIMER 2.0 (http://timer.comp-genomics.org/ (accessed on 28 April 2023)). Immune cell infiltration associated with the development of melanoma was compared to data derived from normal skin samples using the GEO dataset GSE114445 and supplemented with samples from carcinoma in situ. Finally, with the help of data regarding the differentially infiltrated immune cells, the immune cell population of interest was identified. Data used in the re-validation of the immune infiltration assay were obtained from SKCM (cutaneous melanoma) in the TCGA (https://portal.gdc.cancer.gov/ (accessed on 28 April 2023)) project level III HTSeq-FPKM RNAseq data format, and immune cell markers were derived from an immune article [41].

### 4.3. Differentially Expressed Gene (DEGs) Selection and Identification of Key Immune-Related Molecules

The original expression matrix profiles of GSE7929, GSE7956 and GSE8401 were obtained via GEOquery and Biobase packages in R language (in R 4.0.3). The median method was used to normalize the expression matrix. Subsequently, the Limma package (in R 4.0.3) was utilized to identify the differential genes between metastatic and non-metastatic melanoma, and corrected *p*-value calculations were performed to obtain |logFC|. Genes with *p*-values < 0.05 and |logFC| > 1.0 were deemed to be DEGs.

Three sets of DGEs were extracted to perform metastatic melanoma intersecting with DGEs in M1- and M2-type macrophages, and they were identified as macrophage immune marker molecules of metastatic melanoma (MMIMs).

### 4.4. Functional and KEGG Enrichment Analysis of DEGs

The functional enrichment of DEGs was divided into three categories of gene ontology (GO) domain: biological process (BP), cellular component (CC) and molecular function (MF). The KEGG database contains pathway datasets involving biological functions, diseases, chemicals and drugs. In this investigation, significantly upregulated and downregulated DEGs combined with melanoma microarray data were analyzed via R langue (cluster profile package [version 3.14.3] and Org.hs.eg. DB package [version 3.10.0] (for ID conversion) [42]. The detailed data were demonstrated using ggplot2 package (in R 4.0.3).

KEGG and GO analysis were used to characterize important pathway changes in the GSE7929, GSE7956 and GSE8401 of melanoma metastasis. The intersection of metastatic genes of melanoma with the DGEs in M1- and M2-type macrophages was also further analyzed to enable the characterization of critical pathways.

### 4.5. Lasso Method to Determine the Risk Model and Test

The Lasso (least absolute shrinkage and selection operator) method is a compression estimation. It obtains a more refined model by constructing a penalty function that makes it compress some coefficients while setting some coefficients to zero. Therefore, it retains the advantages of subset shrinkage and is a kind of biased estimation used to deal with data with complex co-linearity issues, which can achieve the selection of variables while estimating parameters and better solve the problem of multiple co-linearity in regression analysis.

STAR-count data and the clinical information of SKCM were downloaded from the TCGA dataset repository (https://portal.gdc.com (accessed on 2 May 2023)), from which we extracted data in the TPM format, followed by normalization log2(TPM + 1), and, finally, retained samples with RNAseq data and clinical information, resulting in 471 melanoma patient samples used to perform subsequent analyses.

The optimal model was selected as the final model via first multifactorial cox regression analysis, followed by iterative analysis via a step function. The least absolute shrinkage and selection operator (LASSO) regression algorithm was used to perform feature selection with a 10-fold cross-validation, and the R software glmnet package (in R 4.0.3) was used to perform the above analysis. Based on the LASSO analysis, we constructed prognostic models modeling 167 patients with metastatic melanoma and 471 patients with melanoma in the TCGA database using MMIMs, respectively.

Log rank was used to test KM survival analysis by comparing survival differences between the above more groups, and time ROC analysis was performed to discriminate the accuracy of the prediction model. For Kaplan–Meier curves, *p*-values and hazard ratios (HR) with 95% confidence intervals (CI) were derived via log rank test and univariate Cox regression. All of the above analytical methods and R packages were performed using R software version v4.0.3. *p* < 0.05 was considered to be statistically significant.

### 4.6. Group Comparison of Risk Models

The decision curve (DCA) for a certain overall survival was used to compare the prognostic value of each model and compare these models. We analyzed the DCA curves of the different models for the two models obtained via the Lasso method to determine the characteristic molecules of 167 patients with metastatic melanoma, and all of the above analysis methods and R packages were performed using the v4.1.3 version of R software (R Foundation for Statistical Computing, 2022). *p* < 0.05 was considered to be statistically significant.

### 4.7. Re-Validation of the Model

Using the previous RNAseq data derived from 167 metastatic melanoma patients (level 3) and the corresponding clinical information, univariate and multivariate cox regression analyses were performed, and each variable (*p*-value, HR and 95% CI) was displayed using forest plots via the “forestplot” package. Based on the results of multivariate Cox proportional risk analysis, column line plots were created using the “rms” package to predict the total recurrence rate at the 1-, 2-, and 3-year points. The line graphs provide the graphical results of these factors, allowing the prognostic risk of individual patients to be calculated using the points associated with each risk factor.

### 4.8. Immunocorrelation Analysis of Model

The correlation of the predicted mode with the species immune cells was further analyzed using RNAseq data derived from 167 previous metastatic melanomas (level 3) and the corresponding clinical information. Multi-gene correlation maps were presented via the R package pheatmap. Spearman’s correlation analysis was used to describe correlations between quantitative variables without normal distributions. A *p*-value of less than 0.05 was considered to be statistically significant. All of the above analyses were implemented via R v4.0.3.

### 4.9. TF-Gene-miRNA Interaction Network Based on MMIMs

Genes constituting the model were mapped to the corresponding miRNAs and TFs that controlled the DEGs at a transcriptional level via NetworkAnalyst 3.0 (https://www.networkanalyst.ca/ (accessed on 15 May 2023)), which is a visual online platform that helps to find gene interactions in gene regulatory networks. Comprehensive experimentally validated miRNA–gene interaction data were collected from miRTarBase (15 May 2023). The transcription factor and gene target data were derived from the ENCODE ChIP-seq data [43]. Only the peak intensity signal < 500 and the predicted regulatory potential score < 1 were adopted (using the BETA Minus algorithm http://cistrome.org/BETA/ (accessed on 15 May 2023). The literature-curated regulatory interaction information was collected from the RegNetwork repository http://www.regnetworkweb.org/ (accessed on 16 May 2023)). The ultimate interaction network will be shown on the minimum network.

### 4.10. The Pro-Drug Interaction Network

DrugBank (www.drugbank.ca, (accessed on 16 May 2023)) is a web-based database containing comprehensive information about the influence of numerous drugs on multiple levels, including metabolite, gene expression and protein expression levels [44]. To identify potential drugs that may interact with the genes constituting our model, we performed protein–drug interaction analysis using the DrugBank database (Version 5.0) via NetworkAnalyst. To determine whether related clinical trials reported these potential drugs’ efficacy for melanoma, the drugs were input into the ClinicalTrials.gov registry (https://clinicaltrials.gov/, (accessed on 16 May 2023)) and PharmSnap (https://pharmsnap.zhihuiya.com/, (accessed on 16 May 2023)), which are two highly used and widely trusted worldwide sources of new drugs and drug trials.

## 5. Conclusions

In summary, we identified M1-type macrophages playing important roles in the development and metastasis of melanoma via immuno-infiltration analysis. Based on these results, we narrowed down our focus to 10 genes (NMI, SNTB2, SLC1A4, PDE4B, CLEC2B, IFI27, COL1A2, MAF, LAMP3 and CCDC69) related to the differential expression of M1 and M2 macrophages during melanoma metastasis, with this study representing the first time that such an experiment took place. The immune-related prognostic model based on these 10 constructed genes served as a prognostic predictor of increased immune cell infiltration and metastatic cancer in melanoma. This study identified new melanoma immune-related genes and provided new potential prognostic and therapeutic biomarkers.

## Figures and Tables

**Figure 1 ijms-24-13752-f001:**
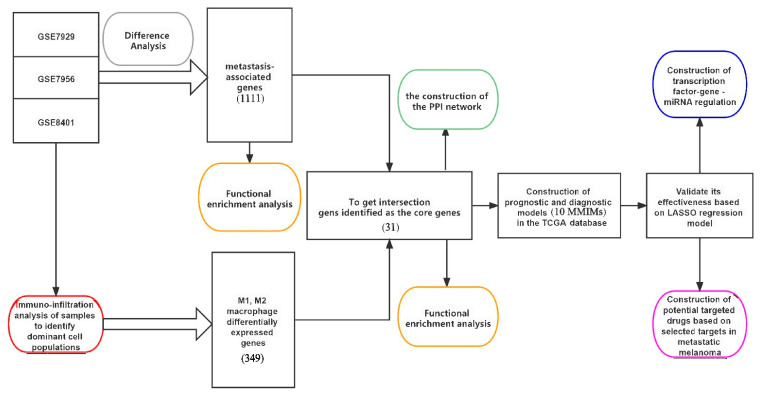
Analyze and define the process of modeling. Different color blocks represent different analytical processes.

**Figure 2 ijms-24-13752-f002:**
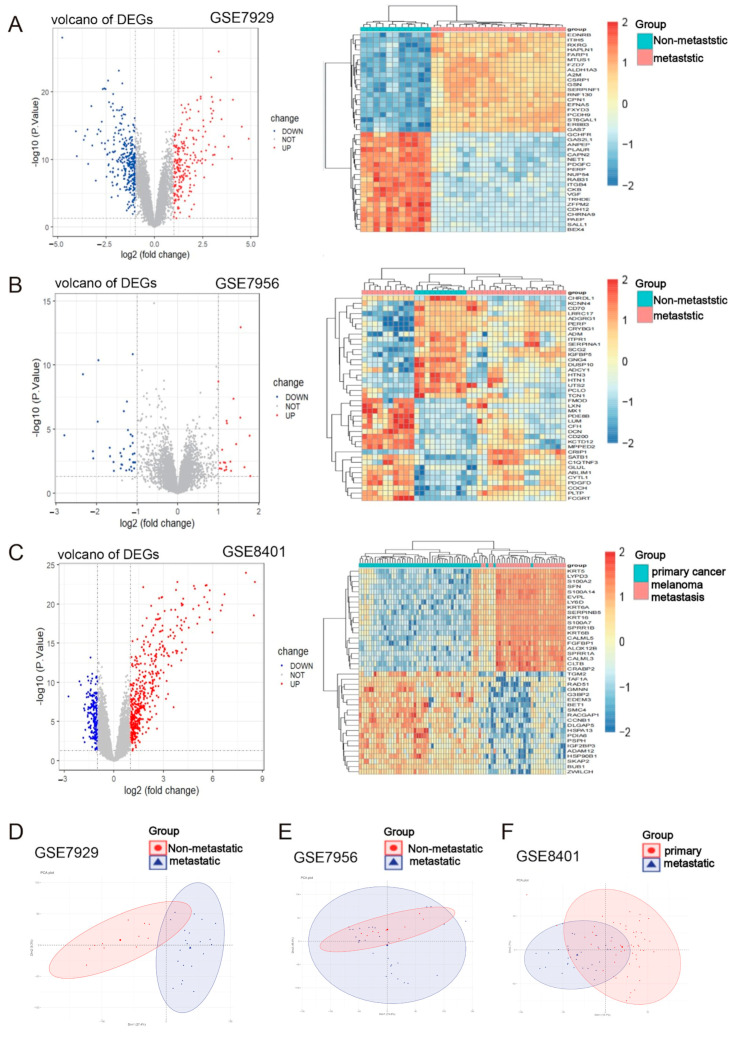
DEGs between the poorly metastatic melanoma group and the highly metastatic melanoma group. (**A**–**C**) The heat map and volcano plot were produced using R software (v4.0.3). The heat map and volcano plot show the expression of DEGs in melanoma in GSE7929, GSE7956 and GSE8401, respectively. The red dots in the graph represent upregulated genes, and the blue dots represent downregulated genes, which are statistically significant in the volcano plot. The gray dots represent genes that are not differentially expressed. The shift in color from green to red represents the progression from low to high expression on the heat map. (**D**–**F**) PCA plots show the compositional differences of each group in the microarray dataset. The red dots represent the expression distribution of metastatic melanoma samples, and the blue triangles represent the expression distribution of non-metastatic melanoma samples.

**Figure 3 ijms-24-13752-f003:**
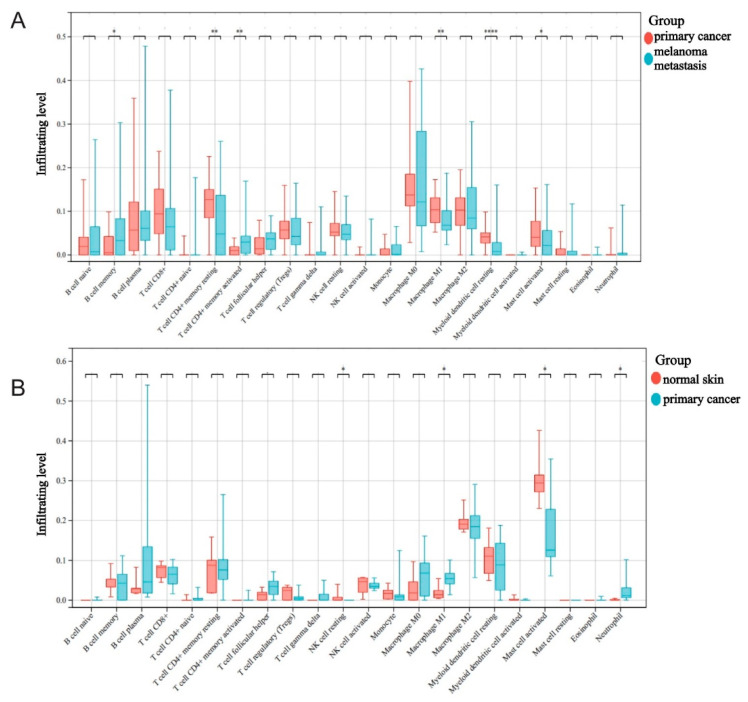
Analysis of immune infiltration in melanoma samples. (**A**) Infiltration of 22 immune cells in melanoma in situ compared to normal skin (data from GSEb114445). (**B**) Comparison between infiltration in 22 immune cells in tumors of patients with melanoma in situ and metastatic melanoma (data from GSE8401, Kruskal–Wallis Test, * *p* < 0.05, ** *p* < 0.01, **** *p* < 0.0001).

**Figure 4 ijms-24-13752-f004:**
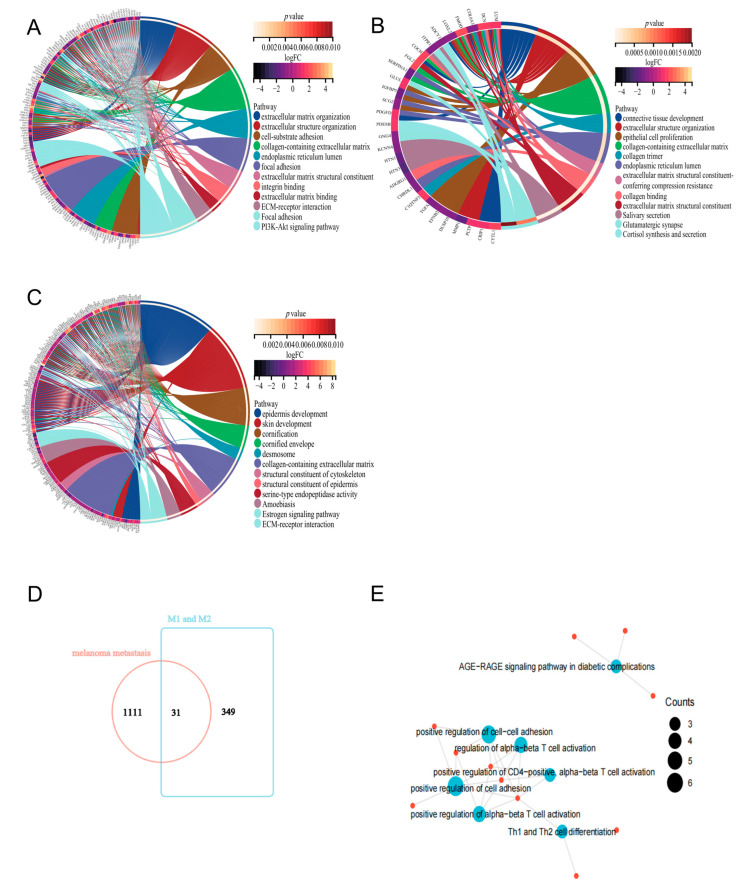
Functional enrichment analysis of genes involved in melanoma metastasis. (**A**) Gene ontology analysis and KEGG pathways of the genes involved in melanoma metastasis in GSE7929. (**B**) Gene ontology analysis and KEGG pathways for the genes involved in melanoma metastasis in GSE7956. (**C**) Gene ontology analysis and KEGG pathways of the genes involved in melanoma metastasis in GSE8401. The outer circle is a bar graph, in which the left half of the bar shows the genes associated with metastasis. The orange and purple shades in the color block on the upper right side represent the *p* and logFC values of the included genes. The under color dots represented the enriched pathway. The inner ring shows the expression distribution of metastasis-related genes that are differentially expressed in each enriched gene ontology term. (**D**) Metastasis-associated melanoma DEGs with M1- and M2-type DEGs were used to intersect. (**E**) Functional enrichment analysis of 31 MMIMs of metastatic melanoma was performed.

**Figure 5 ijms-24-13752-f005:**
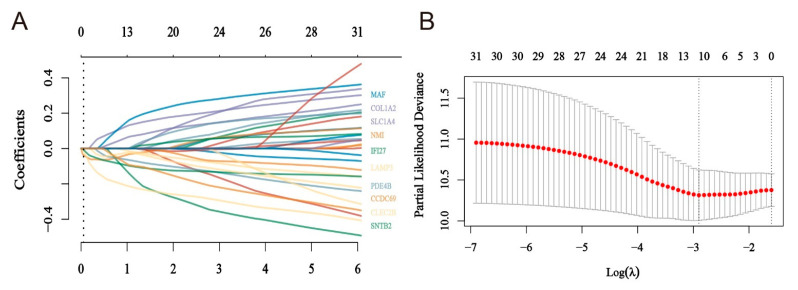
The construction of the model based on MMIMs. (**A**) The construction and validation of the MMIM signature model. The different colored lines represent the distribution of the coefficients of the 31 MMIMs in the regression, marked are the MMIMs selected to build the model. (**B**) The optimal λ was determined when the partial likelihood deviance reached the minimum value. (**C**) The distribution of the risk score, status of the patient and the expression levels of genes in the MMIMs in each patient with melanoma metastatic cancer in the TCGA database. (**D**) The survival curves of patients with low and high risk scores. Solid lines are survival curves and dashed lines are the corresponding 95% confidence intervals. (**E**) The receiver operating characteristic curve (ROC) analysis-predicted overall survival with a risk score.

**Figure 6 ijms-24-13752-f006:**
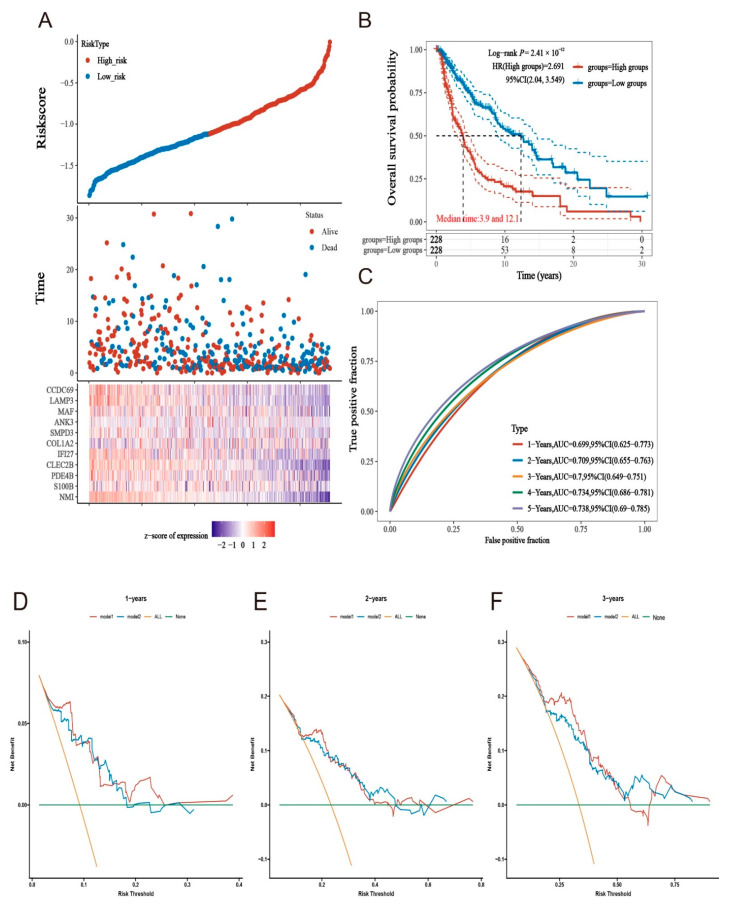
Comparison between the different models used to create constructed models of MMIMs. (**A**) The distribution of the risk score, the status of patient and the expression levels of genes in the MMIM Model 2 in total melanoma patients in TCGA database. (**B**) The survival curves for the low- and high-risk groups. Solid lines are survival curves and dashed lines are the corresponding 95% confidence intervals. (**C**) The ROC analysis predicted overall survival with a risk score. (**D**–**F**) For the DCA decision curves of the two models after 1, 2 and 3 years, the more the curve tends to be in the upper right corner, the better effect. Model 1 was constructed using 31 MMIMs (10 genes) in patients with metastatic melanoma in the TCGA database. Using Model 2, 31 MMIMs were constructed using the total melanoma patients in the TCGA database, and the model was constructed using other genes.

**Figure 7 ijms-24-13752-f007:**
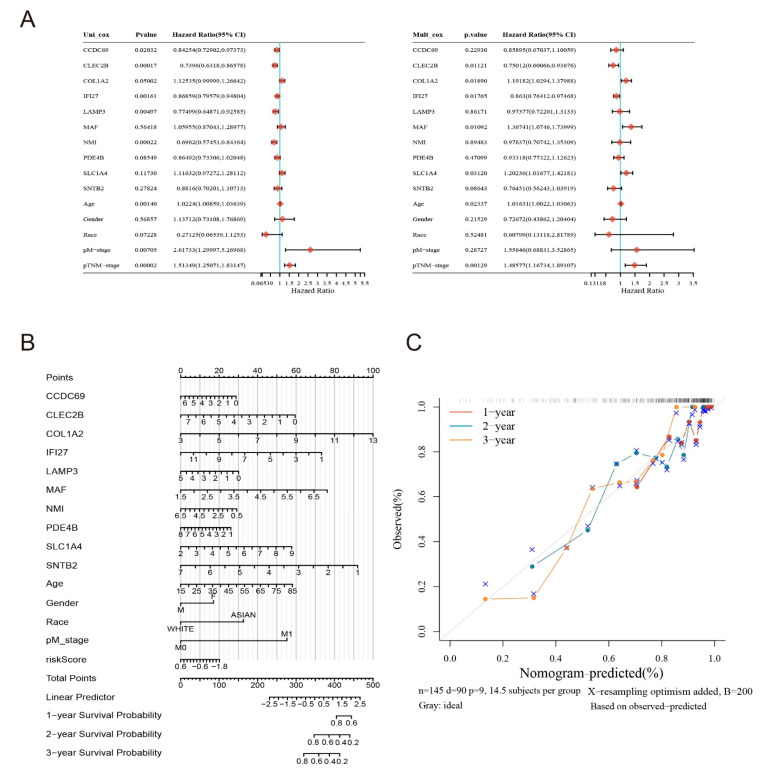
Re-validation of the constructed model. (**A**) Univariate and multivariate analyses of overall survival in melanoma patients in the TCGA databases. (**B**) The predicted 1-, 2- and 3-year survival rates of melanoma patients based on the prognostic nomogram constructed with a risk score and clinicopathological parameters. (**C**) The concordance between predicted and actual 1-, 2- and 3-year survival rates of patients in calibration curves.

**Figure 8 ijms-24-13752-f008:**
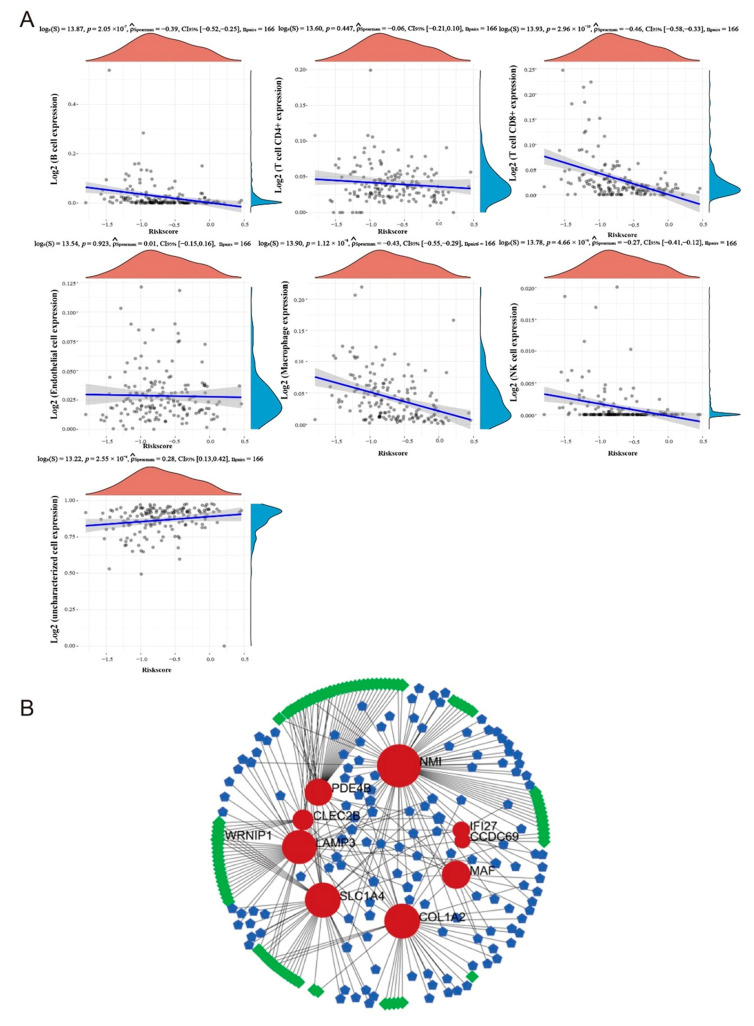
Mechanism correlation analysis of Model 1. (**A**) Relationships between the risk score model and infiltration abundances of seven types of immune cells. (**B**) The main regulatory network constructed based on the genes used in Model 1. The red circular represents MMIMs, the green diamond represent transcription factors and the blue pentagon represents the predicted miRNAs.

**Table 1 ijms-24-13752-t001:** Specific information on the source of the data.

Source Database	Data Type	Data Number Information	Data Grouping Information
GEO	Transcriptome information	GSE7929	Poorly metastatic A375 parental cell line (11) and highly metastatic A375 melanoma metastases (21)
GEO	Transcriptome information	GSE7956	Poorly metastatic A375 parental cell line (10) and highly metastatic A375 melanoma metastases (29)
GEO	Transcriptome information	GSE8401	Clinical primary melanoma (31) and melanoma metastasis (52)
GEO	Transcriptome information	GSE114445	Normal skin (5) and primary cutaneous melanoma (16)
TCGA	Transcriptome information	SKCM	471 patients with melanoma, and 167 of those patients had metastases
MSigDB	Gene Set Collection	GSE11864_1087_200_DN and GSE5099_3808_200_UP	349 differentially expressed genes in M1- and M2-type macrophages

**Table 2 ijms-24-13752-t002:** MMIM information in the TF–gene–miRNA co-regulatory network.

Id	Label (Genes)	Degree	Betweenness
6645	NMI	104	30,695.18
4094	COL1A2	56	15,718.36
26112	SLC1A4	56	15,309.81
6509	LAMP3	51	17,738.41
1278	MAF	30	9543.11
9111	PDE4B	27	9550.66
5142	CLEC2B	13	4370.97
27074	IFI27	8	1965.45
3429	CCDC69	6	1620
9976	SNTB2	4	545.04

**Table 3 ijms-24-13752-t003:** miRNA information in the TF–gene–miRNA co-regulatory network.

Id	Label (miRNAs)	Degree	Betweenness
MIMAT0000083	hsa-mir-26b-5p	4	3041.04
MIMAT0000081	hsa-mir-25-3p	2	6248.14
MIMAT0022720	hsa-mir-1304-3p	2	2174.33
MIMAT0027681	hsa-mir-6890-3p	2	2174.33
MIMAT0000449	hsa-mir-146a-5p	2	1926
MIMAT0000086	hsa-mir-29a-3p	2	1813.46
MIMAT0000765	hsa-mir-335-5p	2	1382.67
MIMAT0000422	hsa-mir-124-3p	2	1295.13
MIMAT0001532	hsa-mir-448	2	1295.13
MIMAT0004597	hsa-mir-140-3p	2	1295.13

**Table 4 ijms-24-13752-t004:** TF information in the TF–gene–miRNA co-regulatory network.

Id	Label (TFs)	Degree	Betweenness
56897	WRNIP1	3	3779.71
171017	ZNF384	2	7894.6
5978	REST	2	2185.42
55929	DMAP1	2	2185.42
467	ATF3	2	2185.42
10346	TRIM22	2	2174.33
51385	ZNF589	2	1553.79
121340	SP7	2	1553.79
865	CBFB	2	1382.67
1999	ELF3	2	1295.13

**Table 5 ijms-24-13752-t005:** Predicted potential drugs.

ID	Molecule and Drug	Drug Trial Phase	Clinical Drug Types	Indications
5142	PDE4B			
DB00131	Adenosine monophosphate	Clinical phase 2/3 merged into clinical phase 3	AMP-activated protein kinase agonist	Type 2 diabetes and metabolic diseases
DB00201	Caffeine	Approved	ADORA2A antagonist and ADORA1 antagonist	Respiratory disease and mental fatigue
DB00277	Theophylline	Approved	Adenosine family receptor antagonist	Immune system diseases and respiratory system diseases
DB00651	Dyphylline	Approved	cDMP-PDE inhibitor	Asthma and chronic bronchitis
DB00806	Pentoxifylline	Approved	Calcium nervous system inhibitor	Cardiovascular disease and type 1 peripheral arterial occlusion xing
DB00824	Enprofylline	-	-	-
DB00920	Ketotifen	Clinical phase 2	-	Immune system diseases, allergic conjunctivitis
DB01088	Iloprost	Approved	Cycloprostenol targeting drug	Respiratory diseases and cardiovascular diseases
DB01113	Papaverine	Approved	cGMP-PDE inhibitor	Neurological diseases and cardiovascular diseases
DB01412	Theobromine	Approved	-	Cardiovascular diseases
DB01427	Amrinone	Approved	cGMP-PDE inhibitor	Cardiovascular diseases
DB01647	*(R)*-Mesopram	Clinical phase II termination	PDE4 inhibitor	Inflammation, multiple sclerosis
DB01656	Roflumilast	Approved	PDE4 inhibitor	Psoriasis, seborrheic dermatitis
DB01791	Piclamilast	Clinical phase 2 termination	PDE4 inhibitor	Asthma and dermatitis
DB01954	Rolipram	Clinical phase 3 termination	PDE4 inhibitor	Neurological diseases and immune diseases
DB01959	3,5-Dimethyl-1-(3-nitrophenyl)-*1H*-pyrazole-4-carboxylic acid ethyl ester	-	-	-
DB02660	Filaminast	Clinical phase 2 termination	Phospholipase A2 inhibitor	Immune system diseases and respiratory system diseases
DB03349	8-Bromo-adenosine-5′-monophosphate	-	-	-
DB03606	*(S)*-Rolipram	Clinical phase 1	PDE4 inhibitor	Huntington’s chorea
DB03807	1-(2-Chlorophenyl)-3,5-dimethyl-*1H*-pyrazole-4-carboxylic acid ethyl ester	-	-	-
DB03849	Cilomilast	Clinical phase 3 Termination	PDE4 inhibitor	Asthma and dermatitis
DB04149	*(R)*-Rolipram	Clinical phase 1	Positron emission tomography enhancer	Neurological diseases
DB04530	S,S-(2-Hydroxyethyl)thiocysteine	-	-	-
DB05219	AN2728	Approved	PDE4 inhibitor	Atopic dermatitis, psoriasis
DB05266	Ibudilast	Approved	PDE4 inhibitor	Uveal melanoma, keratoblastoma, allergic conjunctivitis
DB05676	Apremilast	Approved	PDE4 inhibitor	psoriasis and novel coronavirus pneumonia
DB06909	1-Ethyl-*N*-(phenylmethyl)-4-(tetrahydro-*2H*-pyran-4-ylamino)-*1H*-pyrazolo[3,4-b]pyridine-5-carboxamide	-	-	-
DB08299	4-[8-(3-Nitrophenyl)-1,7-naphthyridin-6-yl]benzoic acid	-	-	-

## Data Availability

The data and code that support the findings of this study are available upon reasonable request from the corresponding authors.

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
