# Peer review of "Development of a Macrophage-Related Risk Model for Metastatic Melanoma"

_ijms, 2023, doi:10.3390/ijms241813752_

Round 1

Reviewer 1 Report

Li et al. present in the paper a correlation of transcriptome information and melanoma-mestases based on 10 genes which are considered to be differentially expressed in M1 or M2 macrophages. The topic is certainly interesting and offers new insights into the disease as well as new targets and methods for the therapy of melanoma patients.

Yet, the following questions need to be addressed by the authors:

1. In the title the authors state that the 10 genes are "macrophage-based". It is not clear what this statement means. Are the molecules expressed by macrophages or not? The authors need to demonstrate e.g. by qPCR that M1 and M2 express these genes. At least some of the genes like CCDC69 have not been reported before to be expressed in macrophages.

2. The selection process of the 10 genes is missing in the paper. It is somehow surprising that the panel does not include any of the standard markers for M1 or M2 macrophages.

3. The text of the manuscript needs to be shortened and could be much more concise. For instance, the Abstract is too voluminous and there are many repetitions e.g. in the Results as well as in the Discussion.

Reviewer 2 Report

The paper integrates different models and methods to fing the relation between metastases formation and other tumor microenvironment cells, mainly macrophages. In my opinion manuscript is interesting, however with moderate neovelty. Manuscript indicate that 10 macrophages related genes can predict metasases formation. Results are supported by numerous bioinformatics analyses. For me as a biologist, the results are interesting and indicate the importance of the tumor microenvironment in the process of metastasis.

The work is very well written and describes the results in an accessible way.

However, the work should be slightly improved.

1. In the introduction section information about M1 and M2 macrophages should be added. A few sentences should be added about the pro-cancer properties of M2 macrophages and anti-cancer M1 macrophages. The markers that these cells are characterized by should be given, which is important in the results shown later.

2. In the discussion section information about 10 sellected molecules should be mentioned. At least their full names and main properies of this proteins. This should make it easier to pinpoint the importance of these particles in the metastasis process.

3. Some language corrections should improve the quality of the manusrcipt. A few that I was able to find: (1) line 49 - anti-CTLA-4 should be, (2) line 66 - tyrosinase, (3) line 66-68 - Why are some markers uppercase and some lowercase?, (4) line 153 - due to its anti-tumor properties.

4. Figure 3 should be of better quality, especially figure 4D, but others in this figure should also have better reslotutions to increase the readability of the text.

5. Figure 7 A is completely unreadable. The text is too small.

The work is very well written. Hovewer, some language corrections should improve the quality of the manusrcipt. A few that I was able to find: (1) line 49 - anti-CTLA-4 should be, (2) line 66 - tyrosinase, (3) line 66-68 - Why are some markers uppercase and some lowercase?, (4) line 153 - due to its anti-tumor properties.

Reviewer 3 Report

9 August 2023

Ms. Ref. No.: ijms-2556976

Journal: International Journal of Molecular Sciences.

Title: An immune-based model of ten M1 and M2 macrophage-based 2 differentially expressed molecule signs for predicting meta- 3 static melanoma prognosis and immune response

Comments:

Thank you for your efforts in writing this article on a very pertinent topic. Moreover, I found the article to be informative and with the potential for further research on this topic in future.

I have some observations where mentioned in the following paragraphs that will be useful for its improvement:

1-      Could you please review the manuscript once more for any grammar errors?

2-      Additionally, there seem to be mistakes in the caption of Figure 4, such as typographical and spelling errors.

3-      It would be helpful to relocate the methods and results section in the manuscript.

4-      Lastly, I was wondering if you could provide the total number of patients whose information was included in the study, as well as the total number of patients that were deemed acceptable for the final study process?

(For example in section 4.5. Lasso method to determine the risk model and test= 471 melanoma patient OR4.6. Group comparison of risk models= 167 patients OR in section 3. Discussion = 102 metastatic melanoma samples and 52 non-metastatic melanoma samples). How can introduce these different sample sizes?

Thanks

Could you please review the manuscript once more for any grammar errors?

Round 2

Reviewer 1 Report

The revised manuscript is now suitable for publication in IJMS.